# Assembly principles of a unique cage formed by hexameric and decameric *E. coli* proteins

Hélène Malet[1,2,3†], Kaiyin Liu[4†], Majida El Bakkouri[4], Sze Wah Samuel Chan[4], Gregory Effantin[2,3], Maria Bacia[5,6,7], Walid A Houry[4*], Irina Gutsche[2,3*]

[1]European Molecular Biology Laboratory, Grenoble, France; [2]Unit for Virus Host-Cell Interactions, Université Grenoble Alpes, Grenoble, France; [3]Unit for Virus Host-Cell Interactions, CNRS, Grenoble, France; [4]Department of Biochemistry, University of Toronto, Toronto, Canada; [5]Université Grenoble Alpes, Institut de Biologie Structurale, Grenoble, France; [6]Institut de Biologie Structurale, CNRS, Grenoble, France; [7]Institut de Biologie Structurale, CEA, Grenoble, France

**Abstract** A 3.3 MDa macromolecular cage between two *Escherichia coli* proteins with seemingly incompatible symmetries–the hexameric AAA+ ATPase RavA and the decameric inducible lysine decarboxylase LdcI–is reconstructed by cryo-electron microscopy to 11 Å resolution. Combined with a 7.5 Å resolution reconstruction of the minimal complex between LdcI and the LdcI-binding domain of RavA, and the previously solved crystal structures of the individual components, this work enables to build a reliable pseudoatomic model of this unusual architecture and to identify conformational rearrangements and specific elements essential for complex formation. The design of the cage created via lateral interactions between five RavA rings is unique for the diverse AAA+ ATPase superfamily.

**\*For correspondence:** walid. houry@utoronto.ca (WAH); gutsche@embl.fr (IG)

[†]These authors contributed equally to this work

**Competing interests:** The authors declare that no competing interests exist.

**Reviewing editor**: Sjors HW Scheres, Medical Research Council Laboratory of Molecular Biology, United Kingdom

## Introduction

Virtually every aspect of cellular function relies on a AAA+ ATPase machine as a key player (*Erzberger and Berger, 2006*; *Snider et al., 2008*). The name of this superfamily, 'ATPase Associated with Diverse Cellular Activities', reflects this remarkable versatility. Widespread among bacteria and archea, members of the MoxR family of AAA+ ATPases are important modulators of multiple stress tolerance pathways (*Snider et al., 2006*; *Wong and Houry, 2012*). The best characterized MoxR representative, the RavA protein, has been recently associated to oxidative stress, antibiotic resistance and iron-sulfur cluster assembly in *Escherichia coli* (*Babu et al., 2014*; *Wong et al., 2014*). Furthermore, it is involved in acid stress and nutrient stress responses via its interaction with the acid stress-inducible lysine decarboxylase LdcI, which buffers the bacterial cytoplasm by transforming lysine into cadaverine while consuming intracellular protons and producing $CO_2$ (*Sabo et al., 1974*; *Soksawatmaekhin et al., 2004*; *Snider et al., 2006*; *Wong and Houry, 2012*). We recently demonstrated that RavA prevented binding of LdcI to its potent inhibitor, the stringent response alarmone, ppGpp (*El Bakkouri et al., 2010*). Thus the LdcI–RavA complex maintains LdcI activity even if the bacterium experiences both acid stress and starvation, as it is often the case in the host stomach through which enterobacteria transit before reaching the bowel where the pathogenesis typically occurs (*El Bakkouri et al., 2010*).

The crystal structures of both individual components of the complex provided important insights into their function (*El Bakkouri et al., 2010*; *Kanjee et al., 2011*). However, the structural principles of the LdcI–RavA interaction are puzzling. Indeed, RavA is a 300 kDa lily-shaped hexameric

**eLife digest** Bacteria inhabit most habitats on Earth, ranging from our bodies to the deepest depths of oceans. In order to thrive in these diverse surroundings, bacteria have developed sophisticated systems that enable them to adapt to changes in their environment. For instance, the bacteria that live in our stomach are exposed to acidic conditions which would normally kill other living organisms, so they have evolved an 'acid stress response' to protect themselves.

A variety of systems are responsible for the acid stress response and in *E. coli*, the most common bacterium in our body, one of these systems relies on two proteins: RavA is a protein that is thought to help other proteins to assemble correctly, while LdcI decreases acidity. These two proteins bind to each other in order to carry out their function.

It is known from previous work that RavA is a symmetric ring-shaped structure made of six equal parts (that is, it is a hexamer), whereas LdcI is made up of two stacked rings, each composed of five equal parts (that is, it is a double pentamer). But how can a hexamer fit together with a double pentamer? This puzzle, which is encountered throughout Nature, is known as 'symmetry mismatch'.

Malet et al. have now used a technique known as cryo-electron microscopy to work out how RavA and LdcI fit together. These experiments show that the part of RavA that makes contacts with LdcI re-arranges itself to form a structure that matches the fivefold symmetry of LdcI. The result is a novel cage-like structure composed by two double pentamers of LdcI, which are parallel to each other and linked together by five hexamers of RavA. Malet et al. propose that this unique structure protects specific proteins from getting destroyed by acids.

ring (*El Bakkouri et al., 2010*), whereas LdcI is a 800 kDa toroid composed of two pentameric rings stacked together back-to-back (*Kanjee et al., 2011*). But how can a hexamer bind a double pentamer? The strength of the interaction ($K_d$ of about 20 nM) between these two symmetry mismatched proteins and the resulting mass of 3.3 MDa inferred from analytical ultracentrifugation came out as a surprise (*Snider et al., 2006*; *El Bakkouri et al., 2010*). The ~35 Å resolution of our first negative stain electron microscopy (EM) reconstruction was however so low that the assembly strategy of the complex, seemingly composed of two LdcI decamers and five RavA hexamers, appeared even more enigmatic (*Snider et al., 2006*). We biochemically showed that the foot of RavA, inserted into the discontinuous triple helical bundle protruding as a leg from the AAA+ ATPase core, is necessary and sufficient for LdcI binding, albeit with a three orders of magnitude lower affinity. We therefore called this foot domain LARA for 'LdcI associating domain of RavA' (*El Bakkouri et al., 2010*). Yet, although speculations were tempting, correct interpretation of the structural bases of the LdcI–RavA interaction was impossible. Hence it was imperative to provide higher resolution insights into the design principles of this intriguing architecture.

## Results

Here we present an 11 Å resolution cryo-electron microscopy (cryoEM) map of the LdcI–RavA complex and a 7.5 Å resolution cryoEM map of the LdcI-LARA complex, which together enable unambiguous flexible docking of the crystal structures, finally fitting together the pieces of the jigsaw. The astounding LdcI–RavA assembly is reminiscent of a symmetrical floral pattern, featuring two parallel five-petal blossoms of LdcI festooned by a garland of five RavA lilies. The entire complex possesses the dihedral fivefold symmetry of the lysine decarboxylase and a large central cavity of $3 \times 10^6$ Å$^3$ (*Figure 1*, *Figure 1—figure supplements 1 and 2*; *Video 1*). Upon complex formation, the RavA hexamer loses its sixfold circular symmetry. The compact outward-pointing hub formed by the six AAA+ modules remains virtually unchanged and the ATP binding sites preserved, whereas the legs massively rearrange in order to comply with the dihedral symmetry of the whole assembly (*Figure 2A*, *Figure 2—figure supplement 1*; *Video 2*). Four protomers of each of the five RavA hexamers are involved in the interaction with the LdcI, while the remaining two legs of each hexamer mediate the RavA–RavA interaction at the equator of the complex (*Video 2*).

Specifically, the RavA hexamers morph into two axially superimposed triskelia rotated 180° with respect to each other (*Figure 2B,C*). The triskelion configuration and the loop (amino acids 329–360) at the beginning of the LARA domain allow the optimal positioning of each LARA domain for

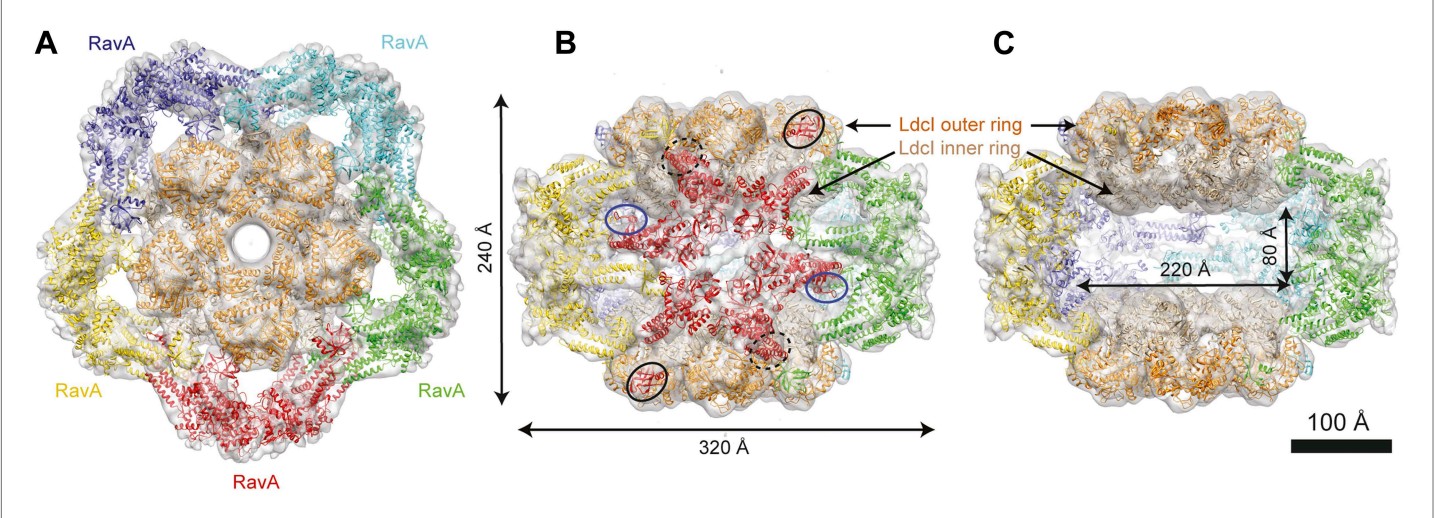

**Figure 1**. Cage-like architecture of the LdcI–RavA complex (11 Å resolution). (**A**) Top view with LdcI facing the reader, (**B**) Side view with RavA facing the reader. For this RavA hexamer, LARA domain positions are indicated by ellipses (solid black for the two LARA domains interacting with the inner LdcI rings from above, dotted black for the two LARA domains interacting with the outer LdcI rings from underneath and invisible from this orientation, solid dark blue for the LARA domains interacting equatorially with the triple helical domains of adjacent RavA monomers). (**C**) Side cut-away view. Complex dimensions are indicated.

The following figure supplements are available for figure 1:

**Figure supplement 1**. Structures of the individual RavA hexamer and the LdcI decamer.

**Figure supplement 2**. CryoEM analysis of the LdcI–RavA cage.

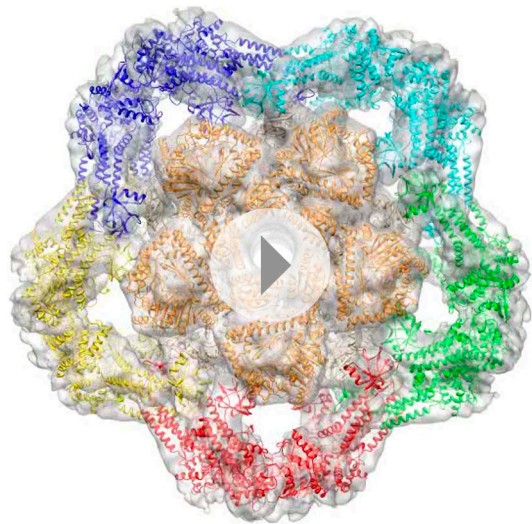

**Video 1**. Overview of the LdcI–RavA cage-like structure.

interaction with LdcI subunits. Indeed, in the context of the complex, the two rings of each LdcI particle are not placed equivalently—one ring of each LdcI faces the exterior of the complex, while the second faces the cavity. Remarkably, one leg of the RavA triskelion stretches its foot out in order to present the LARA domain to the subunits of the inner ring of LdcI from above, while in the second leg the N-terminus of the loop sharply bends, allowing the foot to interact with the outer ring of LdcI from underneath (***Figure 1B***, ***Figure 2B,C,E***, ***Figure 2— figure supplement 1***). As a result of these intricate rotations, both feet appear to interact with the inner or the outer rings of LdcI in exactly the same way.

This surprising finding is confirmed by the 7.5 Å resolution cryoEM map of the LdcI-LARA complex where each LdcI protomer binds one LARA domain, and which can be perfectly superimposed with the 3D reconstruction of the entire LdcI–RavA complex (***Figure 2D***, ***Figure 2—figure supplement 2***, ***Figure 2—figure supplement 3***). At this resolution, the crystal structures of LdcI and LARA can be docked into the EM density unambiguously because the secondary structure elements are clearly resolved (***Figure 2D***, ***Figure 2—figure supplement 3***). The LdcI decamer can be considered as barely affected by the LARA domain binding and all LdcI-LARA interaction sites are indeed equivalent (***Figure 2E,F***,

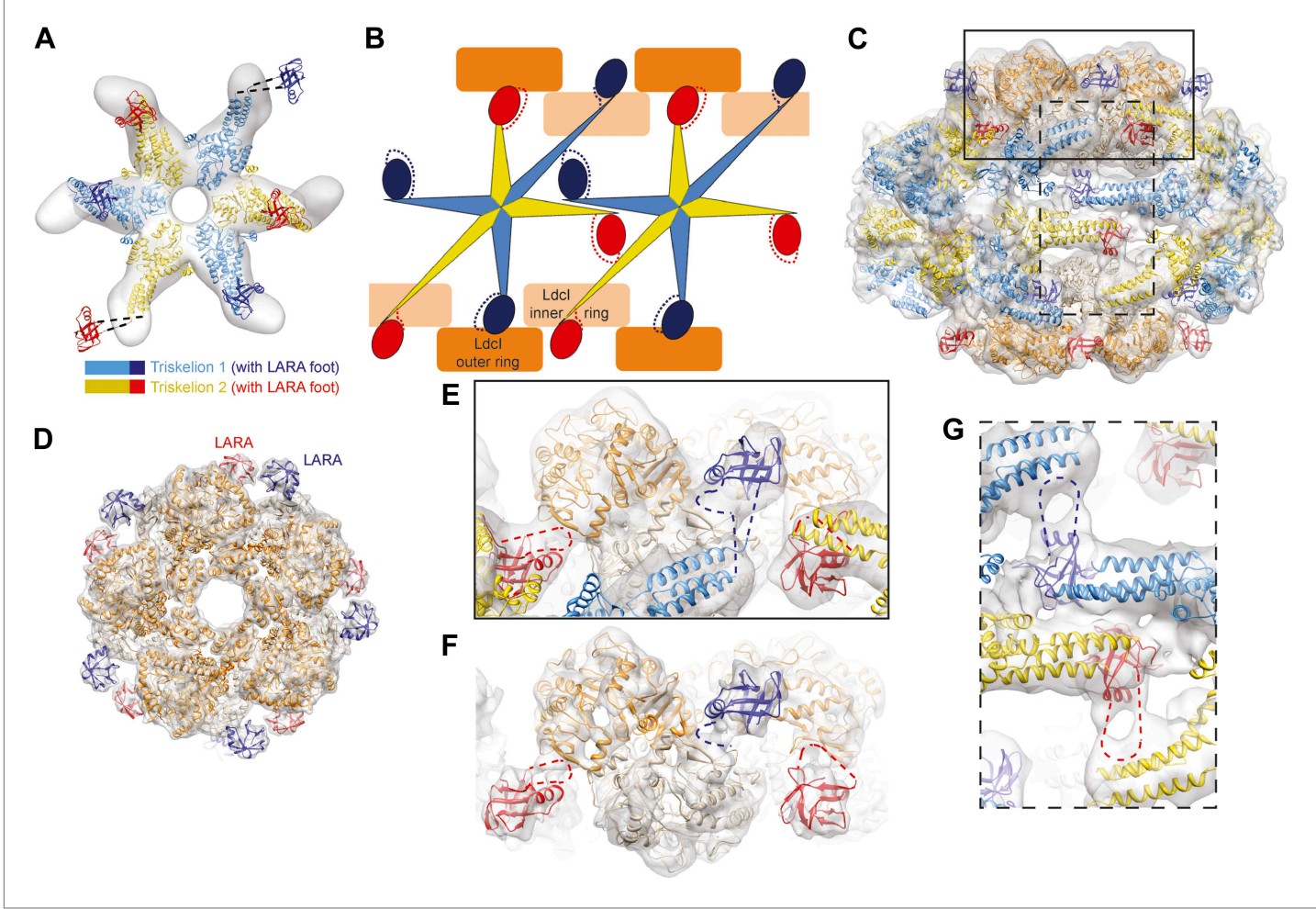

**Figure 2**. The structural organization of the LdcI–RavA cage. (**A**) The RavA hexamer is represented as two triskelia. The pseudoatomic model of the RavA hexamer in the context of the LdcI–RavA complex is superimposed with the isolated RavA negative stain EM map (25 Å resolution, **El Bakkouri et al., 2010**) to show the conformational changes of the RavA legs induced by LdcI binding. (**B–G**) The RavA loop at the beginning of the LARA domain (amino acids 329–360) is shown as a broken line. (**B**) Schematics of LdcI–RavA interaction with RavA. (**C**) CryoEM map and pseudoatomic model of LdcI–RavA (11 Å resolution). This particular orientation of the complex illustrates the origins of the close-up views (**E**) and (**G**) surrounded with a solid rectangle and a broken line rectangle, respectively. (**D**) Top view of the cryoEM map and pseudoatomic model of the LdcI-LARA complex (7.5 Å resolution). (**E**) Close-up view of the LdcI–RavA complex (11 Å resolution) showing the LdcI-LARA interaction and arising from the bold rectangle in (**C**). (**F**) Close-up view of LdcI-LARA (7.5 Å resolution) in the same orientation as in (**E**). The higher resolution of this 3D reconstruction enables a more precise fitting of individual crystal structures. (**G**) Close-up view of the equatorial RavA–RavA interaction via the triple helical bundles and the foot–leg interaction (arising from the broken line rectangle in (**C**)).

The following figure supplements are available for figure 2:

**Figure supplement 1**. Conformation rearrangements of RavA induced by LdcI binding.

**Figure supplement 2**. CryoEM analysis of LdcI-LARA.

**Figure supplement 3**. Insights into the LdcI-LARA interaction in LdcI–RavA and LdcI-LARA maps.

**Figure supplement 4**. Mapping the interaction surfaces between RavA and LdcI.

**Figure supplement 5**. Comparison between triskelia of RavA and clathrin.

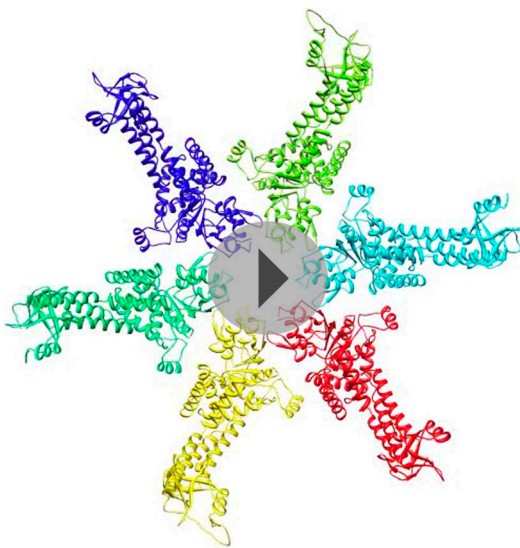

**Video 2**. Sequential representation of the LdcI–RavA complex formation. Morphing of an isolated RavA hexamer into the double triskelion conformation in the context of the LdcI–RavA complex.

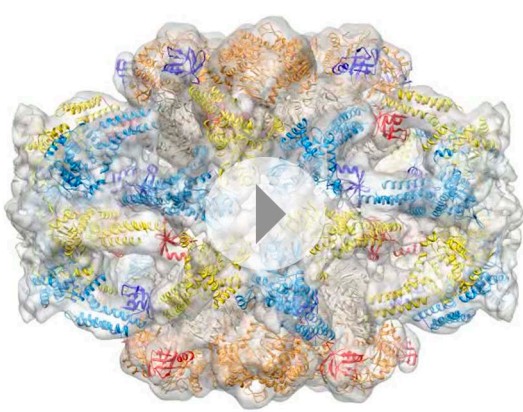

**Video 3**. From the LdcI–RavA cage structure to the interaction between the LdcI and the LARA domains in the higher resolution LdcI-LARA complex. Colors as in *Figure 2*.

*Figure 2—figure supplement 3*; *Video 3*). Noteworthy, it is the loop 329–360 of RavA which is responsible for the LdcI-LARA interaction.

Compared to the minimal LdcI-LARA complex, the scaffold provided by the AAA+ hub and the triple helical bundles of the entire RavA increases the stability of the LdcI–RavA cage. Two legs of each triskelion place the LARA domains on the LdcI. The third legs interact laterally with their counterparts from the neighboring RavA via their triple helical bundles while the above-mentioned loop 329–360 of their LARA domain feet packs against the triple helical domain of a neighboring RavA hexamer, further stabilizing the complex architecture (*Figure 1B*, *Figure 2B,C,G*).

Mutational analysis was carried out to further probe the interaction between LdcI and RavA, and in particular the surface of the LdcI involved in the LARA domain binding, the versatile LARA domain loop and the equatorial interaction between triple helical bundles of RavA. The results are consistent with the resulting pseudoatomic model and identify residues essential for complex stability (*Figure 2—figure supplement 4*). Thus, critical residues in LdcI required for RavA binding were found to be E634 and Y697, both part of a C-terminal β-sheet (*Figure 2—figure supplement 4A,C,D*). The charged residues R347 and R348 and the hydrophobic residues I343 and F344 at the tip of the LARA domain loop are required for RavA binding to LdcI (*Figure 2—figure supplement 4A–D*). Hence, the data suggest that the interaction between the two proteins is mediated both by charged and hydrophobic residues. Moreover, shortening the loop by deleting residues 329–335, or constraining its N-terminus by introducing two prolines (S331P, D332P), weakens or abolishes the LdcI–RavA complex formation (*Figure 2—figure supplement 4A–D*). Finally, mutating residues R315 and E452 in the first and second helices of the RavA triple helical bundle, respectively, destabilized complex formation (*Figure 2—figure supplement 4A,E,F*) suggesting that the RavA–RavA leg–leg interaction is mediated by these two helices.

To summarize, the same loop at the N-terminus of the LARA domain appears involved in (1) positioning the foot in respect to the rest of the leg so that to allow for equivalent interaction with both the inner and the outer LdcI rings, (2) binding to LdcI, and (3) binding to the triple helical bundle of the adjacent RavA (*Figure 2B,E–G*). This loop therefore clearly emerges as the main determinant of the LdcI–RavA cage formation. While the 7.5 Å resolution of the current LdcI-LARA reconstruction precludes building a reliable pseudoatomic model of the 329–360 LARA domain loop based on the experimental EM density, its structural plasticity is obviously essential to fullful this role (*Figure 2—figure supplement 4*).

## Discussion

The triskelion representation of the RavA hexamer inside the LdcI–RavA assembly brings out a parallel between the LdcI–RavA complex and the eukaryotic vesicle coats, in particular clathrin coats (*Harrison*

*and Kirchhausen, 2010*; *Faini et al., 2013*; *Figure 2—figure supplement 5*). Indeed, clathrin is a triskelion-shaped trimer with a compact hub jutting out three alpha-superhelical legs tipped with a domain that interacts with adapters and cargo molecules. Clathrin triskelia assemble in vitro into cages of pentagons and hexagons, the hubs representing the cage vertices and the helical legs intertwining between vertices in a two-fold symmetrical manner to create the edges.In the case of the LdcI–RavA cage, the LARA domain feet appear to be specifically evolved for the LdcI binding (*El Bakkouri et al., 2010*). The equatorial vertices of the cage are provided by the hub made by the AAA+ domains of RavA, whereas the edges are created by the triple helical legs related by a twofold symmetry and intertwining between two adjacent AAA+ domain vertices. The fact that LdcI–RavA complex and eukaryotic vesicle coats have certain architectural principles in common is striking, because the LdcI–RavA architecture is remarkably different from all AAA+ ATPase assemblies described so far. Indeed, to our knowledge it is the only complex (out of 30 released EM maps in EMDB and 43 oligomeric crystal structures in PDB in July 2014) composed of several laterally interacting AAA+ ATPase rings, and the only one enclosing a central cavity other than the cavity framed by the AAA+ modules assembled into a ring or a spiral.

Why would the cell create such an exquisite and elaborate architecture simply to prevent ppGpp interaction with LdcI? One may posit an explanation based on the markedly higher stability of the entire LdcI–RavA complex compared to the LdcI-LARA complex: the scaffold provided by the AAA+ hub and the triple helical legs of RavA is required to glue the LARA domain in place and effectively preclude ppGpp binding. Noteworthy, the LARA foot binds ~30 Å away from the closest ppGpp binding pocket. Thus, contrary to our earlier prediction based on the low resolution negative stain EM map (*Snider et al., 2006*; *El Bakkouri et al., 2010*), the LARA domain does not appear to directly block the access of ppGpp to its binding pocket but rather to induce a local conformational change in this pocket reducing ppGpp affinity.

Based on the fact that RavA is a MoxR AAA+ ATPase and that the MoxR family is known to have chaperone-like functions important for maturation or assembly of specific protein complexes, and taken into account the cage design of the LdcI–RavA complex, it would be tempting to suggest that this architecture may also fulfill yet another role. Along these lines, an involvement of RavA and its binding partner, a VWA-domain protein ViaA, in the Fe–S cluster assembly and particular respiratory pathways (*Babu et al., 2014*; *Wong et al., 2014*) and a physical interaction of RavA, ViaA (*Wong et al., 2014*) and LdcI (*Erhardt et al., 2012*) with specific subunits of the highly conserved respiratory complex I (*Erhardt et al., 2012*; *Wong et al., 2014*) were recently documented. Our structure of the exquisite LdcI–RavA cage substantiates a hypothesis that it may act as a chaperone, protecting substrates from denaturation or disassembly under acid stress conditions inside the central cavity, and spurs on further functional investigation. Here we unraveled the layout principles of the unique LdcI–RavA edifice, elucidated conformational rearrangements and specific elements essential for complex formation, and made a step towards general understanding how Nature elegantly solves the problem of the symmetry mismatch between individual components of protein complexes.

## Materials and methods

### Protein purification and interaction studies

RavA mutants were generated via the QuickChange method (Stratagene, La Jolla, California). Proteins were expressed from p11 plasmid (*Zhang et al., 2001*) which encodes an N-terminal 6xHis-tag followed by the Tobacco Etch Virus (TEV) protease cleavage site. Plasmids were transformed into BL21-(DE3) Gold pLysS strain and overexpression induced by addition of 0.5 mM Isopropyl β-D-1-thiogalactopyranoside (IPTG) at 30°C for 4 hr. All RavA mutants were purified as previously described (*El Bakkouri et al., 2010*). Briefly, proteins were first purified by Ni-NTA affinity chromatography, then incubated with TEV protease overnight to remove the N-terminal His-Tag, and further purified via ion exchange chromatography using a MonoS column and by size exclusion chromatography on a Superdex S200 size exclusion column.

LdcI mutants were also generated via the QuickChange method. Wild type and mutant proteins were expressed from pET22b plasmid which encodes a C-terminal uncleavable 24 amino acid linker containing 6xHis-tag. The plasmids were transformed into CF1693 strain bearing the pT7-322-Tet[r] plasmid to express the T7 polymerase. CF1693 contains a deletion of *relA* and *spoT* genes, which renders the strain not capable of generating ppGpp (*Xiao et al., 1991*). Protein overexpression was

induced by addition of 1 mM Isopropyl β-D-1-thiogalactopyranoside (IPTG) at 18°C for 20 hr. Proteins were purified on Ni-NTA resin, followed by ion exchange chromatography using a MonoQ column, and then by size exclusion chromatography on a Superdex S200 column.

The isolated LARA domain was expressed and purified as described (*El Bakkouri et al., 2010*).

The rationale behind the mutants was provided by sequence alignment between three species of enterobacteria with the closest LdcI and RavA—*Escherichia coli*, *Salmonella enterica* and *Enterobacter aerogenes*—and by the pseudoatomic models produced in this study. For example, the fit of the crystal structures of the LdcI and the LARA domain of RavA into the LdcI-LARA reconstruction clearly shows that the β-strands 632–636 and 696–698 of the LdcI are involved in the interaction with the loop 329–360 of the LARA domain. In the case of the LdcI, both β-strands being strictly conserved between the three bacterial species compared, a charged (E634) and a hydrophobic (Y697) residue were chosen to probe the interaction and mutated into an oppositely charged (E634K) and a polar uncharged (Y697S) residue. Mutating the LdcI residues D638 and E681, situated further away from the indicated β-sheet did not affect the RavA–LdcI interaction (data not shown). In the case of the LARA domain of RavA, the loop 329–360 could not be modeled based on the current resolution of the EM map. Moreover, this loop appeared to be the main determinant of the LdcI–RavA interaction and to undergo major conformational rearrangements in comparison to the crystal structures of the RavA monomer (see the main text and the methods below). Therefore the length of the loop (deletion of amino acids 329–335), the flexibility of its N-terminus (S331P, D332P), as well as four residues in the middle of the loop which might be involved in the interaction (two charged, R347 and R348, and two hydrophobic, I343 and F344) were chosen for mutations (*Figure 2—figure supplement 4*). The four latter residues are shown only in the *Figure 2—figure supplement 4A* but not in the *Figure 2—figure supplement 4D* to avoid overinterpretation of the fit provided that their precise position cannot be predicted.

To test complex formation between the various LdcI and RavA mutants, gel filtration chromatography was performed on a Superose 6 column. LdcI and RavA proteins were mixed at a ratio of 2:3 and incubated in a buffer containing 25 mM TrisHCl pH 7.9, 200 mM NaCl, 5% glycerol, 10 mM MgCl$_2$, 0.1 mM PLP, 2 mM ATP, and 1 mM DTT for 30 min prior to injection onto the column. Protein samples were collected in 1 ml fractions and resolved on SDS-PAGE gels. All gels were then silver stained.

## LdcI-LARA complex—cryoEM data collection and 3D reconstruction

For LdcI-LARA complex formation, 0.6 mg/ml of LdcI was mixed with 1 mg/ml of LARA for 15 min at room temperature in a buffer containing 50 mM Mes pH 6.5, 100 mM NaCl, 0.2 mM PLP, 1 mM DTT and 0.01% glutaraldehyde (glutarahaldehyde initially buffed in 800 mM Mes pH 6.5). The LdcI:LARA ratio was 1:10 (7.3 µM of LdcI for 73 µM of LARA). The glutaraldehyde cross-linking reaction was stopped by adding a final concentration of 62 mM TrisHCl pH 7. 4 µl of sample was applied to glow-discharged quantifoil grids (Quantifoil Micro Tools GmbH, Germany) 400 mesh 2/1, excess solution was blotted during 2 s with a Vitrobot (FEI) and the grid frozen in liquid ethane (*Dubochet et al., 1988*). Data collection was performed on a FEI Polara microscope operated at 300 kV under low dose conditions. CryoEM micrographs were collected on a CCD Ultrascan Gatan USC 4000 (4 k × 4 k) using FEI EPU automatization software at a magnification of 102.413×, giving a pixel size of 1.464 Å. The contrast transfer function (CTF) for each micrograph was determined with CTFFIND3 (*Mindell and Grigorieff, 2003*) and corrected by phase flipping using bctf (*Heymann et al., 2008*). Defocus ranged between 1.5 and 2.7 µm.

For initial 3D model determination, a subset of 11,102 particles was picked manually using EMAN boxer routine (*Ludtke et al., 1999*). Band-pass filtered particles were centered against a rotationally averaged total sum and classified using multivariate statistical analysis (MSA) as implemented in IMAGIC (*van Heel et al., 1996*). A subset of ~10 class averages was selected (based on the visual match between the class average and the individual particles) as references for multi-reference alignment (MRA). After three rounds of MRA/MSA, a 3D model was calculated by angular reconstitution. As two and fivefold symmetries were clearly visible in class averages and eigenimages, D5 symmetry was imposed for 3D calculations. Particle orientations were refined by multiple cycles of MRA, MSA and angular reconstitution, gradually incorporating more particles. The resulting initial model of LdcI-LARA (~18 Å resolution) was used for particle picking using an automated particle picking procedure based on the Fast Projection Matching algorithm (*Estrozi and Navaza, 2008*).

The resulting dataset of 26,165 LdcI-LARA particles was then used to refine the initial model by projection matching (AP SH command in SPIDER, D5 symmetry imposed) (*Frank et al., 1996*; *Shaikh et al., 2008*). When the alignments had stabilized, more than 95% of the images aligned to the same references in consecutive rounds of alignment. The final 3D reconstruction comprised 23,540 particles. The resolution was estimated based on the gold-standard FSC = 0.143 criterion (*Scheres and Chen, 2012*) by dividing the data in two independent halves and refining them iteratively against the angular reconstitution model low pass filtered to 50 Å resolution. The resolution estimated based on the unmasked FSC curve was 8.8 Å, whereas the FSC curve obtained with soft shaped masks (dilated 5 pixels and with a fall–off profile of a cosine half-bell of 4 pixel width) yielded the resolution of 7.5 Å (*Figure 2—figure supplement 2D*). The final map was sharpened with EMBfactor (*Fernández et al., 2008*) using calculated B-factor of 463 Å$^2$.

## LdcI–RavA complex—cryoEM data collection and 3D reconstruction

Conditions of optimal LdcI–RavA complex formation were initially tested by negative stain EM on a JEOL 1200 EX microscope. Different pHs (ranging from 6 to 8), salt concentrations (ranging from 100 to 300 mM), protein ratios (ranging from 1:1.5 to 1:3 LdcI monomer:RavA monomer), ADP concentrations (from 0 to 5 mM) were tested. No cross-linking agent was used to stabilize this high affinity complex. The most promising conditions were further analyzed by cryoEM on a FEI CM200 microscope. The final optimal condition chosen for data collection was: 1.26 mg/ml of LdcI mixed with 0.94 mg/ml of RavA for 15 min at room temperature in a buffer containing 25 mM Mes pH 6.5, 200 mM NaCl, 3 mM ADP, 0.8 mM PLP and 1 mM DTT. The resulting ratio of LdcI:RavA was 1:2 (in the complex, the ratio is LdcI–RavA ratio is 1:1.5, so there was 1.3 molar excess of RavA). Quantifoil grids were flash frozen in liquid ethane (*Dubochet et al., 1988*) using Vitrobot and a blotting time of 2 to 3 s.

Initial 3D model was determined using cryo-electron tomography (cryoET) and subtomogram averaging as follows. Tomograms were recorded on a FEI Polara microscope between −65 and 65° with 2° angular step on a CCD Ultrascan Gatan USC 4000 at a nominal magnification of 51160× giving a pixel size of 2.932 Å. The total electron dose for each tomogram was around 40 electrons/Å$^2$. The tomograms were aligned with the IMOD suite using 5 nm gold fiducial beads (BBInternational, UK) for frames alignment. 95 subvolumes were extracted in IMOD and aligned using PEET (*Kremer et al., 1996*). The subvolume average was consistent with the previously determined negative stain EM map of LdcI–RavA (*Snider et al., 2006*; *Figure 1—figure supplement 2*) and clearly displayed a D5 symmetry.

A high resolution dataset was then collected on a FEI Polara microscope operated at 300 kV under low dose conditions. Micrographs were recorded on Kodak SO-163 film at 59,000 magnification, with defocus ranging from 1.3 to 3.3 µm. Films were digitized on a Zeiss scanner (Photoscan) at a step size of 7 µm giving a pixel size of 1.186 Å. After CTF determination and correction (performed as for the LdcI-LARA dataset), particles were automatically picked using the Fast Projection Matching algorithm with projections of the cryoET reconstruction as a template. The resulting dataset of 35,443 particles was then used to refine the initial cryoET model by projection matching in SPIDER with D5 symmetry imposed (*Frank et al., 1996*; *Shaikh et al., 2008*). 60% of the best correlating particles were used to refine the 3D reconstruction. When the alignments had stabilized, more than 95% of the images aligned to the same references in consecutive rounds of alignment. The final map comprised 21,265 particles with an even view distribution around the equatorial axis. The resolution was estimated based on the gold-standard FSC = 0.143 criterion (*Scheres and Chen, 2012*) by dividing the data in two independent halves and refining them iteratively against the cryoET model low pass filtered to 50 Å resolution. The resolution estimated based on the unmasked FSC curve was 14 Å, whereas the FSC curve obtained with soft shaped masks (dilated 5 pixels and with a fall–off profile of a cosine half-bell of 4 pixel width) yielded the resolution of 11 Å (*Figure 1—figure supplement 2D*).

## Fitting of atomic models into cryoEM maps

For LdcI-LARA fitting, LdcI decamer crystal structure (pdb 3N75) was fitted into the LdcI-LARA cryoEM map using the fit-in-map module of UCSF Chimera (*Pettersen et al., 2004*), unambiguously identifying extra densities corresponding to LARA domains. The density of the LARA domain (residues 361–424, pdb 3NBX) was extracted and the LARA atomic model reliably rigidly fitted using SITUS (*Wriggers, 2012*). Subtle domain movements being noted in the LdcI structure as compared to the crystal structure

(*Figure 2—figure supplement 3D,E*), the LdcI density was extracted and used for flexible fitting with Flex-EM (*Topf et al., 2008*). LdcI crystal structure was divided into three rigid bodies: the wing domain (residues 1–128), an intermediate domain (residues 131–142, 145–382, 386–397, 402–495, 527–562) and a C-terminal domain (residues 501–521 and 566–710), leaving inter-domains flexible. The Cα RMSD between the initial pdb of the LdcI monomer model and the final Flex-EM model is 0.65 Å. The cross correlation values between the cryoEM map and either the rigidly fitted monomer or the final Flex-EM model are 0.75 and 0.84 respectively. The FSC curves between the cryoEM map and either the rigid fit or the Flex-EM model give an FSC = 0.5 at 11 Å resolution and 8.3 Å resolution respectively (*Figure 2—figure supplement 3F*).

For LdcI–RavA fitting, superposition of LdcI–RavA and LdcI-LARA maps (*Figure 2—figure supplement 3A–C*), demonstrated that LdcI and LARA positions were conserved in both maps (cross-correlation of 0.86 between the two maps filtered at 11 Å resolution). In order to take advantage of the higher resolution of the LdcI-LARA map, the LdcI-LARA fit was kept fixed for the LdcI–RavA fitting. Thus, only the AAA+ ATPase domain and the triple helical domains of RavA (pdb 3NBX) remained to be fitted. The initial rigid fit of these domains using UCSF Chimera fit-in-map module indicated that the AAA+ ATPase domain position was conserved compared to the isolated RavA structure, maintaining the integrity of the ATPase active sites. The rigid fit also revealed that RavA binding to LdcI required movements of the triple-helix domains to bring the LARA domains in their LdcI-interacting position, whereas the RavA legs involved in RavA–RavA equatorial interactions barely moved in comparison to the isolated RavA (*Figure 2—figure supplement 1*). In order to propose a pseudoatomic model of RavA in its LdcI-binding state, RavA flexible fitting was thus performed using Flex-EM. The ATPase domains (residues 1–268), the triple helical bundles (residues 274–329, 444–496) and the LARA domains (residues 361–424) were considered as rigid, leaving inter-domains flexible. The LARA domains were given as initial position their position in LdcI-LARA maps, and the initial positions of the AAA+ ATPase domains were those of the rigid body fit. 25 cycles of Flex-EM led to improved position of triple-helix domains leaving the AAA+ ATPase and LARA domain fit quasi unchanged. The versatile 330–360 loop of the LARA domain was omitted from the models because it clearly undergoes major rearrangements that cannot be reasonably modeled based on the current quality of the EM maps.

The FSC curves between the cryoEM map of the LdcI–RavA complex and either the rigid fit of the LdcI decamer and the RavA hexamer, or the Flex-EM model obtained as described, give an FSC = 0.5 at 26 Å resolution and 20 Å resolution respectively (*Figure 2—figure supplement 3G*). Interestingly, the FSC curve between the LdcI-LARA part of the LdcI–RavA map and its Flex-EM model has an FSC = 0.5 at 11 Å resolution, whereas the FSC curve between the equatorial region of the map (corresponding to the AAA+ domain core and the triple helical domains of RavA) and its Flex-EM model has an FSC = 0.5 at 20 Å resolution. This difference in the local quality of the Flex-EM model can be explained by (i) the better quality of the Flex-EM model of the LdcI-LARA region (because this part of the model is based on the higher resolution LdcI-LARA map), and (ii) the better definition of the LdcI-LARA region of the LdcI–RavA map in comparison to the equatorial region corresponding to the AAA+ domain core and the triple helical domains of RavA.

## Accession codes

CryoEM maps and Cα traces of the corresponding fitted atomic structures have been deposited in the Electron Microscopy Data Bank and Protein Data Bank, respectively, with accession codes EMD-2679 and PDB-4upb for LdcI–RavA, EMD-2681 and PDB-4upf for LdcI-LARA.

## Acknowledgements

HM and IG thank Guy Schoehn and Stephen Cusack for support. For electron microscopy, this work used the platforms of the Grenoble Instruct center (ISBG; UMS 3518 CNRS-CEA-UJF-EMBL) with support from FRISBI (ANR-10-INSB-05-02) and GRAL (ANR-10-LABX-49-01) within the Grenoble Partnership for Structural Biology (PSB). The electron microscope facility (Polara electron microscope) is supported by the Rhône-Alpes Region (CIBLE and FEDER), the FRM, the CNRS, the University of Grenoble and the GIS-IBISA. HM was supported by a long-term EMBO fellowship (ALTF413-2011). KL was the recipient of the Ontario Graduate Scholarship and the Life Sciences Award from the University of Toronto. MEB was the recipient of a fellowship from the Canadian Institutes of Health Research (CIHR) Strategic Training Program in Protein Folding and Interaction Dynamics: Principles and Diseases.

SWSC was the recipient of the National Sciences and Engineering Research Council of Canada (NSERC) Undergraduate Student Research Award. This work was supported by the French ANR-12-JSV8-0002 to IG and by a grant from the CIHR (MOP-130374) to WAH.

## Additional information

### Funding

| Funder | Grant reference number | Author |
|---|---|---|
| Agence Nationale de la Recherche | ANR-12-JSV8-0002 | Irina Gutsche |
| Canadian Institutes of Health Research | MOP-130374 | Walid A Houry |
| European Molecular Biology Organization | ALTF413-2011 | Hélène Malet |
| Ontario Ministry of Training, Colleges and Universities | Ontario Graduate Scholarship | Kaiyin Liu |
| University of Toronto | Life Sciences Award | Kaiyin Liu |
| Canadian Institutes of Health Research | | Majida El Bakkourri |
| National Sciences and Engineering Research Council of Canada | | Sze Wah Samuel Chan |

The funders had no role in study design, data collection and interpretation, or the decision to submit the work for publication.

### Author contributions

HM, Structural characterization of the samples, Electron microscopy data collection and analysis, Drafting the manuscript; KL, MEB, SWSC, Design of mutants, Purification and biochemical characterization of samples; GE, Structural characterization of the RavA hexamer; MB, Assistance with electron microscopy data collection; WAH, Conception and direction of the biological studies, Contribution to drafting the manuscript; IG, Electron microscopy data analysis, Conception and direction of the structural studies, Drafting the manuscript

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
