## [Decision Letter]

Thank you for sending your work entitled “Assembly principles of the unique cage formed by the AAA+ ATPase RavA hexamer and the lysine decarboxylase LdcI decamer” for consideration at *eLife.* Your article has been favorably evaluated by John Kuriyan (Senior editor) and 3 reviewers, one of whom is a member of our Board of Reviewing Editors.

The following individuals responsible for the peer review of your submission have agreed to reveal their identity: Sjors Scheres (Reviewing editor).

The Reviewing editor and the other reviewers discussed their comments before we reached this decision, and the Reviewing editor has assembled the following comments to help you prepare a revised submission.

All reviewers found the symmetry mismatch and the way Nature solved this “problem” in your structure intriguing, and agreed that the paper should therefore be published in *eLife.* However, all reviewers also agreed that more context about the function of the complex should be provided in a revised version of the manuscript. One reviewer suggested that, apart from providing a more extensive introduction on the biological function of this complex, you should also test the described mutants in a stress response assay. Although such data may indeed strengthen the manuscript (and we would therefore encourage you to include such data if available), a consensus was reached that such data are not a necessary requirement for publication of the current manuscript.

The following points would need to be addressed in a revised manuscript:

1) In the main text, the introductory comments, as well as the description of some of the results and the final discussion would benefit from some expansion in order to make the paper more accessible to the general reader. As also mentioned above, more context about the function of this complex should be provided in the introductory paragraphs. In addition, it would be insightful to describe in somewhat more detail in what aspects the clathrin coat resembles your structure. If necessary, you may relax the word-count constraints of the short format to implement these changes.

2) The authors do not mention the use of independently refined half-models to estimate their resolution, and report their resolution using the FSC=0.5 criterion. It is now commonly accepted in the EM-field that such a procedure is dangerous as it potentially overfits the data. It is therefore recommended that the authors adapt their SPIDER scripts so that 2 models are refined independently. The FSC curve between the resulting models could then be interpreted using the FSC=0.143 criterion.

3) It would be insightful to obtain more quantitive information about the quality of the fitted atomic models. In particular, the authors should present FSC curves between the fitted atomic models and the 2 cryo-EM maps. As flexible-fitting may suffer from overfitting, it would be good to present 2 curves for each map: one versus the rigid-body fitted models and one versus the flexibly fit models.

---

## [Author Response]

*All reviewers found the symmetry mismatch and the way Nature solved this “problem” in your structure intriguing, and agreed that the paper should therefore be published in eLife. However, all reviewers also agreed that more context about the function of the complex should be provided in a revised version of the manuscript*.

Opting for a short report format, we had previously decided to concentrate on the main findings which are the structural bases of the LdcI-RavA interaction. We are now happy to add functional aspects in several places in the manuscript (see below).

*One reviewer suggested that, apart from providing a more extensive introduction on the biological function of this complex, you should also test the described mutants in a stress response assay. Although such data may indeed strengthen the manuscript (and we would therefore encourage you to include such data if available), a consensus was reached that such data are not a necessary requirement for publication of the current manuscript*.

Although the particular mutants described in this study were only tested *in vitro*, we know from our previous work (El Bakkoury et al., PNAS 2010) that the wild type *E.coli* strain increases the pH of its growth media at a higher rate than the *ravA-deltaLARA* strain expressing a RavA variant devoid of the LARA domain (i.e. incapable of interaction with LdcI).Therefore we strongly suppose that the present RavA mutants that don’t bind LdcI would also have a similar effect on the acid stress *in vivo*.

*1) In the main text, the introductory comments, as well as the description of some of the results and the final discussion would benefit from some expansion in order to make the paper more accessible to the general reader. As also mentioned above, more context about the function of this complex should be provided in the introductory paragraphs. In addition, it would be insightful to describe in somewhat more detail in what aspects the clathrin coat resembles your structure. If necessary, you may relax the word-count constraints of the short format to implement these changes*.

In this version, we fully rewrite the introductory paragraphs and the Discussion to include all available functional data and implications of our structural findings in the light of the most recent biological discoveries. In addition, the comparison with clathrin coats is extended.

*2) The authors do not mention the use of independently refined half-models to estimate their resolution, and report their resolution using the FSC=0.5 criterion. It is now commonly accepted in the EM-field that such a procedure is dangerous as it potentially overfits the data. It is therefore recommended that the authors adapt their SPIDER scripts so that 2 models are refined independently. The FSC curve between the resulting models could then be interpreted using the FSC=0.143 criterion*.

In our alignment and reconstruction procedure we already paid a great attention not to overfit the data: for example, we low pass filtered the 3D map and the particle images before each new refinement cycle, gradually increasing the resolution upon successive refinement iterations, and stopped as soon as typical overfitting “artefacts” started to appear. However, anticipating this request of the reviewers, we now also split the data in two independent halves and processed it separately. In this revised version, we provide the FSC curves between the two halves of the data sets calculated with unmasked and soft shaped masks (as in Chen et al., Ultramicroscopy 2013), so that the reader can appreciate the quality of the processing (Figure 1—figure supplement 2, Figure 2—figure supplement 2).

*3) It would be insightful to obtain more quantitive information about the quality of the fitted atomic models. In particular, the authors should present FSC curves between the fitted atomic models and the 2 cryo-EM maps. As flexible-fitting may suffer from overfitting, it would be good to present 2 curves for each map: one versus the rigid-body fitted models and one versus the flexibly fit models*.

To alleviate potential overfitting distortions at the medium resolution of our EM maps and considering that the crystal structures of the LdcI and the RavA monomers are composed of three distinct domains each, we left the individual domains rigid and allowed flexibility only for short interdomain loops in order to place each domain correctly into the EM density (see Materials and Methods).

The requested curves are now presented in Figure 2—figure supplement 3 and described in the Materials and Methods section.